# Possible Role of Carbocysteine Syrup in the Deflation of Percutaneous Endoscopic Gastrostomy Balloons

**Gabrio Bassotti** [1,*] and **Danilo Castellani** [2]

1   Gastroenterology, Hepatology, and Digestive Endoscopy Section, Department of Medicine and Surgery, University of Perugia, 06123 Perugia, Italy
2   Gastroenterology Unit, Perugia General Hospital, 06129 Perugia, Italy; danilo.castellani@ospedale.perugia.it
*   Correspondence: gabassot@tin.it

**Abstract:** Percutaneous endoscopic gastrostomy is the method of choice to allow enteral access in patients requiring long-term enteral nutrition. However, although generally safe, percutaneous tube positioning may be plagued by several complications. Among these, the deterioration and/or deflation of balloons serving as internal bolster is particularly worrisome in that it may lead to gastrostomy cannulas dislocation. Of interest, such balloon deflation may occur in up to 30% of cases for apparently unexplained causes. Here, we provide a hypothesis that could explain some of these causes.

**Keywords:** balloon deflation; carbocysteine syrup; drug administration; PEG cannula

## 1. Introduction

To date, enteral and parenteral feeding represent important ways to provide nutritional support to meet metabolic requirements in subjects unable to assume food by oral route [1,2]. In patients with an adequate function of the gastrointestinal tract, enteral feeding is preferred to parenteral feeding. The latter, in fact, carries a substantial risk of infections, is more costly, and cannot provide enteric stimulation, thus compromising the defensive barrier of the gut [1].

Since its introduction in the clinical arena in the 1980s, percutaneous endoscopic gastrostomy (PEG) has become the modality of choice for providing intestinal access to patients with a functional gastrointestinal tract who require long-term enteral nutrition, usually beyond four weeks [1–3]. In addition, there is evidence that PEG is superior to nasogastric tube feeding in improving nutrition and preventing common complications for patients with swallowing disturbances [4]. Although generally considered safe, PEG tube placement can be associated with many potential complications, classified as major or minor [5]. Major complications are represented by aspiration pneumonia, bleeding, buried bumper syndrome, bowel perforation, necrotizing fasciitis, and metastatic seeding [1,5]. Minor complications, representing the majority of PEG complications, include peristomal wound infection, granuloma formation, tube leakage into the abdominal cavity causing peritonitis, stoma leakage, inadvertent PEG removal, tube blockage, pneumoperitoneum, and gastric outlet obstruction [5]. Moreover, another relatively frequent minor complication is due to the deterioration and/or deflation of PEG balloons, which serve as the internal bolster, and whose performance may be disappointing [6]. The deflation of a PEG balloon, in fact, often causes PEG dislocation [1].

It is worth noting that, apart from incorrect positioning and handling, PEG balloon deflation may occur for apparently unexplained causes in up to about 30% of patients [7]. In this paper, we provide a possible explanation for some such instances, likely due to factors that should be known by all those involved in the care of patients with PEG placement.

## 2. Case Report

We observed two geriatric patients, aged respectively 86 and 89 years, with previous tracheostomy positioning, who underwent PEG positioning for neurological causes (stroke and dementia) preventing food ingestion by mouth. After successful positioning, the patients returned several times to our attention in brief (fifteen to thirty days) time intervals for PEG cannula displacement due to balloon deflation. In every instance of these late dislodgments, the positioned balloons were repeatedly and accurately checked, and no leaks or other apparent causes for these repeated deflations (including stomal infections and peristomal leakage) were identified. However, after a few months of several such displacements, on strict pharmacological questioning it emerged that both patients were given carbocysteine lysine salt syrup, 10 mL q.i.d, through the PEG cannula, in order to prevent or decrease tracheal secretions. Since we suspected that the drug could have been involved in the balloon deflations, we discussed the issue with the patients' caregivers and suggested to stop the administration of the carbocysteine syrup. Of interest, after such withdrawal both patients did not have further balloon deflations in the following months, and no PEG cannula displacements were reported at one year follow-up. Therefore, we hypothesized that the mucolytic action of carbocysteine [8] might somewhat influence the permeability of the PEG balloons, leading to their progressive deflation in the time course.

To test this hypothesis, we evaluated the possible role of carbocysteine lysine salt syrup (FLUifort®, Dompè Farmaceutici, Milano, Italy, 90 mg/mL, the same drug administered to our patients) in the deterioration or deflation of PEG balloons. To this purpose, four PEG cannulas commonly used in our practice (A-Kimberly-Clark 14 Fr, B- Kimberly-Clark 16 Fr, C-BARD 16 Fr, and D-HALYARD 22 Fr, all of medical grade silicone construction, natural rubber, latex-free) were tested. The balloons of each PEG were filled at time 0 with distilled water according to the manufacturer's instructions: 5 mL for A and B, 20 mL for C, and 10 mL for D. Thereafter, the balloons were immersed in sterile containers filled with 50% FLUifort® syrup and 50% distilled water (80 mL total), to mimic gastric dilution, and maintained in immersion for four weeks. Four identical PEG cannulas were also tested in the same manner, but with the balloons immersed in 80 mL distilled water. During the four weeks of observation, the balloons were checked and photographed at weekly intervals for possible macroscopic deterioration (breaks, colorimetric changes), and at the end of the observation period the amount of distilled water within the balloons was retrieved and measured.

In the course of the observation period of four weeks, no gross deterioration or visible leaks of the balloons was visually appreciated in both groups of probes. However, as shown in Table 1, at the end of the observation period there was a decrease in the retrieved distilled water content of all balloons. This decrease was of modest entity for balloons immersed in distilled water and of greater extent in those immersed in the solution containing carbocysteine lysine salt syrup (Figure 1).

**Table 1.** Percentage of balloon deflation observed in devices immersed in carbocysteine lysine salt syrup (FLUifort®) and distilled water.

| | PEG Immersed in Solution with FLUifort® | | | | PEG Immersed in Water | | | |
|---|---|---|---|---|---|---|---|---|
| | A | B | C | D | E | F | G | H |
| Time 0 | 5 mL | 5 mL | 20 mL | 10 mL | 5 mL | 5 mL | 20 mL | 10 mL |
| After 4 w | 2 mL | 1.5 mL | 11 mL | 6 mL | 4.5 mL | 4.8 mL | 19 mL | 9 mL |
| Difference | −60% | −70% | −45% | −40% | −10% | −4% | −5% | −10% |
| Average | | −54% | | | | −7% | | |

Identical devices: A and E, B and F, C and G, D and H.

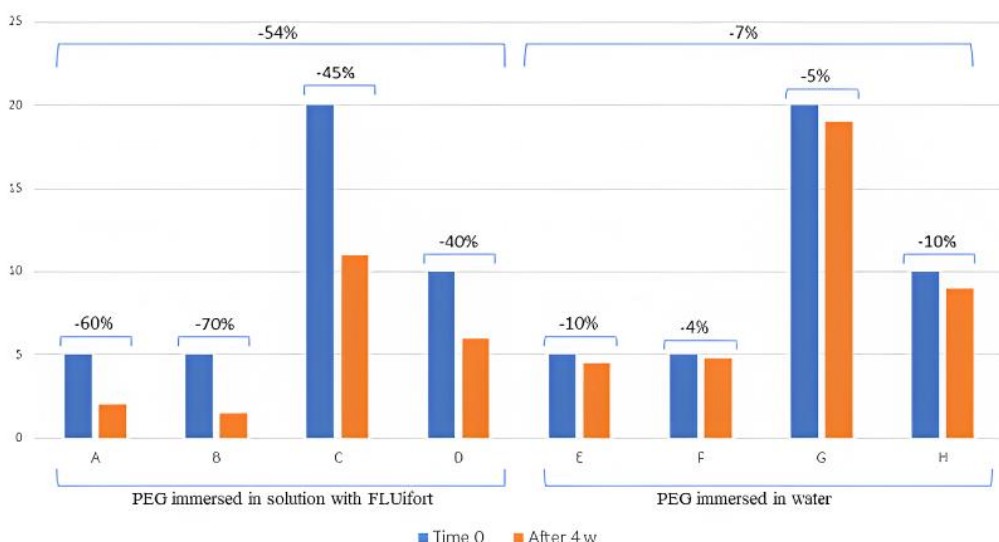

**Figure 1.** Graphic representation of the percentage of balloon deflation in devices immersed in carbocysteine lysine salt syrup (FLUifort®) and distilled water. Identical devices: A and E, B and F, C and G, D and H.

### 3. Discussion

To date, PEG has become the preferred choice for patients needing long-term nutritional support [1–3], and performs better compared to radiological positioning of feeding tubes [9]. Moreover, the replacement of gastrostomy feeding tubes after PEG positioning is a procedure that is easily done at a patient's home, being safe and effective with relevant cost reduction [10]. Notwithstanding its usefulness, it must always be remembered that PEG is an invasive procedure, and that complications may develop. Fortunately, most of these complications are those considered as minor [1,5]. One of these, and among the most frequent, is represented by balloon deflation, which occurs in an unexplained manner in about 30% of patients after PEG positioning [7]. Of note, balloon deflation may subsequently lead to PEG cannula dislodgement and migration due to the loss of the balloon's function as an internal bolster [11]. In this report, we describe an unexpected and unforeseen potential cause of PEG balloon deflation that is likely due to the administration of a mucolytic drug.

Although based on a relatively crude observation, we feel that these findings, coupled to the clinical benefit, might be useful in daily clinical practice and provide some practical advice. In fact, the clinical longstanding benefit we observed after the drug was suspended provides ground for the hypothesis that treating patients with carbocysteine lysine salt syrup through PEG cannulas might account for balloon deflation. The latter could be due to an increase in the balloon's permeability by the drug, due to its capability as penetration enhancer [12,13], since no gross detectable lesions were observed. On the other hand, the PEGs were accurately checked during several occasions to exclude common causes of dislodgment, and no balloon abnormalities, infections, or peristomal leakages were detected. Of course, this hypothesis needs more thorough and specific testing, including the need to be tested on a greater number of different devices. However, due to the almost immediate benefit we observed after drug discontinuation, leading to the interruption of the repeated balloon deflation, we suggest medical caregivers to obtain a thorough drug history of patients undergoing PEG positioning in a daily clinical setting in order to avoid pitfalls and patients' discomfort related to potentially preventable causes.

### 4. Conclusions

PEG cannula positioning is a very useful manner to feed patients unable to ingest food per os. Although generally safe, this technique may cause complications, frequently due to balloon deflation and cannula dislodgments. Here, we showed evidence that carbocysteine

syrup may potentially cause PEG balloon deflation: therefore, from a clinical point of view, it is important to always obtain an accurate pharmacologic history (also on unrelated conditions) in order to avoid, or limit as much as possible, interferences with the patient's quality of life.

**Author Contributions:** Conceptualization: G.B. and D.C.; methodology: G.B. and D.C.; formal analysis: D.C.; investigation: G.B. and D.C.; writing—original draft preparation: G.B.; writing—review and editing: G.B. and D.C. All authors have read and agreed to the published version of the manuscript.

**Funding:** This research received no external funding.

**Institutional Review Board Statement:** Not applicable.

**Informed Consent Statement:** Informed consent was obtained from all subjects involved in the study.

**Data Availability Statement:** No new data were created for this study.

**Conflicts of Interest:** The authors declare no conflict of interest.

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
