# Peer review of "Possible Role of Carbocysteine Syrup in the Deflation of Percutaneous Endoscopic Gastrostomy Balloons"

_clinpract, doi:10.3390/clinpract13020043_

Round 1

Reviewer 1 Report

The study "Possible role of carbocysteine syrup in the deflation of percutaneous endoscopic gastrostomy balloons" presents an interesting clinical observation that discusses the effect of carbocysteine syrup on deflation of gastrostomy balloons. 

This observation is another interesting voice in the discussion on the effect of drugs and decontaminators on the durability of materials from which feeding tubes were made.

However, I believe that for the full clinical usefulness of this study, it is important to supplement 

(A-Kimberly-Clark 14 Fr, B-Kimberly-Clark 16 Fr, C-57 BARD 16 Fr, and D-HALYARD 22 Fr) 

about the material, from which these catheters were made.

Author Response

The study "Possible role of carbocysteine syrup in the deflation of percutaneous endoscopic gastrostomy balloons" presents an interesting clinical observation that discusses the effect of carbocysteine syrup on deflation of gastrostomy balloons. 

This observation is another interesting voice in the discussion on the effect of drugs and decontaminators on the durability of materials from which feeding tubes were made.

However, I believe that for the full clinical usefulness of this study, it is important to supplement 

(A-Kimberly-Clark 14 Fr, B-Kimberly-Clark 16 Fr, C-57 BARD 16 Fr, and D-HALYARD 22 Fr) 

about the material, from which these catheters were made.

RESPONSE: added, as suggested

Reviewer 2 Report

The Authors reported two cases of  deslodgement of PEG tubes in patients receiving a mucolytic agent through the tubes, and they hypotesized the potential role of this agent in increasing the permeability of the ballon and hence its deflation. They did a simple experiment supporting their hypotesis.

The take home message to collect  a thorough medical history in patients undergoing PEG, and esplecially in those with repeated tube dislogements is useful for the clinicians. 

I have non major comments

I only suggest to specify that both cases were "late dislodgements", occuring after 2 weeks from PEG placement (fistula tract present).

Besides, in the introduction (page 1, rows 22-23) they mention minor complications, but not the major ones. I would specifiy which are the major compliactions (necrotizing fascitis, colocutaneous fistula), and I would also outline that most complications are minor.

Author Response

The Authors reported two cases of  deslodgement of PEG tubes in patients receiving a mucolytic agent through the tubes, and they hypotesized the potential role of this agent in increasing the permeability of the ballon and hence its deflation. They did a simple experiment supporting their hypotesis.

The take home message to collect  a thorough medical history in patients undergoing PEG, and esplecially in those with repeated tube dislogements is useful for the clinicians. 

I have non major comments

I only suggest to specify that both cases were "late dislodgements", occuring after 2 weeks from PEG placement (fistula tract present).

Besides, in the introduction (page 1, rows 22-23) they mention minor complications, but not the major ones. I would specifiy which are the major compliactions (necrotizing fascitis, colocutaneous fistula), and I would also outline that most complications are minor.

RESPONSE: specified both issues, as required

Reviewer 3 Report

The authors say that carbocysteine syrup might have a role in the deflation of the PEG balloon. This may be true, because many substances can alter the quality of the balloon, for example, a mycoses infection can create acidity and thus distort deflation. In these two cases, the possibility of stoma infection with mycoses or other agents has not been evaluated.

Author Response

The authors say that carbocysteine syrup might have a role in the deflation of the PEG balloon. This may be true, because many substances can alter the quality of the balloon, for example, a mycoses infection can create acidity and thus distort deflation. In these two cases, the possibility of stoma infection with mycoses or other agents has not been evaluated.

RESPONSE: added this information, previously omitted for brevity

Round 2

Reviewer 3 Report

It is Ok

Author Response

(The authors gave the same response as above.)
